# A Comparison of the Intrarectal and Intramuscular Effects of a Dexmedetomidine, Ketamine and Midazolam Mixture on Tear Production in Cats: A Randomized Controlled Trial

**DOI:** 10.3390/ani14010145

**Published:** 2023-12-31

**Authors:** Andrea Paolini, Massimo Vignoli, Nicola Bernabò, Amanda Bianchi, Roberto Tamburro, Maria Cristina Pincelli, Francesca Del Signore, Andrea De Bonis, Martina Rosto, Francesco Collivignarelli, Clelia Distefano, Ilaria Cerasoli

**Affiliations:** 1Faculty of Veterinary Medicine, University of Teramo, 64100 Teramo, Italy; mvignoli@unite.it (M.V.); nbernabo@unite.it (N.B.); abianchi@unite.it (A.B.); mcpincelli@unite.it (M.C.P.); fdelsignore@unite.it (F.D.S.); adebonis@unite.it (A.D.B.); mrosto@unite.it (M.R.); fcollivignarelli@unite.it (F.C.); clelia.distefano@yahoo.it (C.D.); 2Clinica Veterinaria Borghesiana, 00132 Rome, Italy; ilaria.cerasoli@gmail.com

**Keywords:** intrarectal route, intramuscular route, cats, tear production, dexmedetomidine, ketamine, midazolam, sedation

## Abstract

**Simple Summary:**

In veterinary medicine, the use of sedative agents to perform clinical and diagnostic procedures is increasingly common. Several anesthetic drugs decrease tear flow production. In recent years, interest in the clinical efficacy of sedative and hypnotic agents for new and atraumatic routes of administration has increased. Tear film production has never been investigated in cats who received sedative agents by the intrarectal route (IR). The aim of this study is to compare the clinical effects of a mixture of dexmedetomidine, ketamine and midazolam on tear film flow administered by the IR versus the intramuscular route (IM).

**Abstract:**

Cats are often easily stressed and uncooperative. The use of sedative agents in the feline species is widely used to perform even minor clinical and diagnostic procedures. The aim of this study is to assess the impact on tear film production of the intrarectal route (IR) administration of a mixture of dexmedetomidine, ketamine and midazolam in comparison with the intramuscular (IM) one. A group of twenty cats were involved in a randomized and blinded clinical trial. A clinical and ophthalmological examination was conducted on the cats. The IR group received dexmedetomidine 0.003 mg kg^−1^, ketamine 4 mg kg^−1^ and midazolam 0.4 mg kg^−1^; the IM group received dexmedetomidine 0.003 mg kg^−1^, ketamine 2 mg kg^−1^ and midazolam 0.2 mg kg^−1^. A Shirmer tear test I (STT- I) was conducted 1 h before sedation and 2′, 10′, 20′, 30′, 40′, and 80′ post drug administration. The reaction to STT-I administration was also evaluated. The IM group has a lower mean tear production than the IR group for all time points evaluated. Cats in the IM group showed less reaction to STT-I administration. This study may suggest that the effect of sedative agents administered by the IR route has a lower incidence on tear production than the IM one. The use of eye lubricant is recommended in any case.

## 1. Introduction

In veterinary medicine, sedative drugs are widely used to carry out small clinical diagnostic procedures that would otherwise be difficult to perform on animals who are awake [1]. Despite the recommendations of feline-friendly handling, nursing care and environmental guidelines [2,3,4], sedation in cats is often necessary, as they are poorly cooperative and easily stressed animals [5]. The intramuscular (IM) and intravenous (IV) administration routes are the most used options for sedation in dogs and cats [6], although they have potential side effects. In the literature, the incidence of malignant neo-formations following IM injection is reported in cats [7,8]. The IM injection is considered the most stressful and painful administration route in people [9]. In veterinary medicine, the use of a eutectic cream (a mixture of prilocaine and lidocaine) has been investigated to reduce venipuncture pain in cats [10]. On the other hand, it is not always possible to place a venous catheter due to the lack of patient cooperation. Furthermore, correct management of the venous catheter to prevent different complications such as extravasal administration, inflammation and/or phlebitis is desirable [11,12]. In recent years, in dogs and cats, attention has been paid to the efficacy and safety of sedative and analgesic drugs that are administered by alternative routes such as oral (OS), trans-mucosal (TM), intranasal (IN) and intrarectal (IR) [5,13,14,15,16,17]. For the IR route, few studies on the efficacy and complications of anesthetic and analgesic drugs are published. A recent study confirms the effectiveness and the few side effects of IR administration in cats, including a lower reaction to administration compared to the IM route [17].

Tear secretion maintains an optimal extracellular environment for epithelial cells of the cornea and conjunctiva [18]. Tears also provide a protective system for the ocular surface and antibacterial and antinociceptive activity [19,20]. The lacrimal glands with cholinergic and adrenergic fibers are innervated by the lacrimal nerve, which originates from the trigeminal nerve [19,21]. Therefore, autonomic nervous system interactions can affect tear film production. Tears have two modalities of secretion. The “basal production” is a continuous low-level flow, covering the avascular cornea and helping maintain an adequate homeostatic balance [22]. The other one, “reflex production” is a marked transient secretion caused by mechanical stimuli of the ocular surface. The task of this second form of tear production is aimed at protecting the eye such as by removing foreign bodies or irritants from the ocular surface [22].

A lack of tear production can lead to inflammation of the conjunctiva and the cornea, a pathological condition known as kerato-conjunctivitis sicca [23,24]. The most widely used method for the quantitative assessment of tear production is the Schirmer Tear Test I (STT-I). In human and veterinary medicine, the STT-I quantifies basal and reflex tear production [18,24]. More studies have been conducted regarding the correlation between sedative drugs and the decrease in tear production in dogs and cats [23,25,26], and for this reason, it is recommendable to apply eye lubricant after sedative administration to ensure corneal protection [25]. To the authors’ knowledge, no study has investigated the impact of the IR route on tear film production in cats.

The aims of this study are as follows: (1) Evaluate tear production in sedated cats with dexmedetomidine, ketamine and midazolam after two different routes of administration (IR vs. IM); (2) Compare the reaction to the STT-I placement in these groups. The main hypothesis is that IR administration in cats has a reduced impact on tear production compared to the IM route. The other hypothesis is that the IR and IM groups have the same reaction when performing the STT-I.

## 2. Materials and Methods

### 2.1. Animals

A group of 20 owned domestic shorthair cats (14 males and 6 females) were involved in the study. The cats were admitted to the Veterinary Teaching Hospital “G. Gentile” (University of Teramo) for X-rays or abdominal ultrasound procedures that required chemical immobilization. Owners were notified and asked to sign a written consent form for each cat enrolled. The study was approved by the Committee on Animal Research and Ethics of the University of Chieti-Pescara, Teramo, L’Aquila and the Experimental Zooprophylactic Institute of Abruzzo-Molise (CEISA), protocol n° 8, 3 March 2020. Cats were randomly assigned to the groups using dedicated website (www.random.org), accessed on 10 November 2021. Drug administration was performed by two operators who were not involved in the collected data (ADB and MR). Inclusion criteria were ASA I or II based on clinical examination and execution of routine blood test screenings (hematology and serum biochemistry). Pregnant cats, ophthalmic, rectal and perineal disease were considered exclusion criteria. The cats were fasted 12 h before sedation, and water remained available until the start of the procedure.

### 2.2. Procedure

Control group (IM group) received intramuscular injection of dexmedetomidine (Dextroquillan; Fatro, Bologne, Italy) 0.003 mg kg^−1^, ketamine 2 mg kg^−1^ (Ketavet; MSD Animal Health, Kenilworth, NJ, USA) and midazolam 0.2 mg kg^−1^ (Midazolam; Pharma Hameln, Hamelin, Germany); experimental group (IR group) received endorectal administration of dexmedetomidine 0.003 mg kg^−1^ (Dextroquillan; Fatro, Bologne, Italy), ketamine 4 mg kg^−1^ (Ketavet; MSD Animal Health, Kenilworth, NJ, USA) and midazolam 0.4 mg kg^−1^ (Midazolam; Pharma Hameln GmbH, Hamelin, Germany). The injection in IM group was made on the longissimus dorsi muscle, while the IR group received drug mixture in the rectum with insulin syringe without needle (Micro-fine 1 mL; BD, New York, NY, USA). Regardless of the volumes to be administered, the syringe was advanced gently to 0.3 mL into the rectum. Before IR administration, all syringes were lubricated with sterile water solution. No rectal emptying or enema was executed before drug administration in IR group. To exclude oculist pathologies, the day before the procedure, an ophthalmological examination was conducted by a veterinary ophthalmologist (CP): threat reaction, palpebral reflex, corneal reflex, pupillary light reflex, dazzle test, swinging flashlight test, STT-I, eye staining with fluorescein and intraocular pressure were recorded before procedure started. To perform the STT-I, dedicated ophthalmic strips were used (SCH-100, Eyecare, Allahabab, India). The paper strips were placed firstly in the right eye then in the left, previously bent 90° for one minute within the ventral conjunctival sac of each cat (the cornea was never touched to avoid “reflex production”). Time points were as follows: 1 h before drug administration, (T0) 2′ (T1), 10′ (T2), 20′ (T3), 30′ (T4), 40′ (T5) and 80′ minutes post drug administration (T6). Reaction to STT-I placement was evaluated at the same time points. Reaction score is presented in Table 1.

A 22-gauge intravenous catheter (Jelco; Smiths Medical, Minneapolis, MN, USA) was aseptically inserted into a cephalic vein 15 min after the drug administration. Ringer’s Lactate solution (Baxter Healthcare Corp, Deerfield, IL, USA) was administered at a rate of 3 mL kg^−1^ h^−1^. Flow-by oxygen was available for the entire procedure at 150 mL kg^−1^ min^−1^. Cardiovascular parameters such as heart rate (HR), systolic, mean and diastolic arterial pressure (SAP, MAP and DAP) were recorded by a high definition oscillometry detector (HDO; S+B medVet GmbH, Babenhausen Germany), and respiratory rate (*f*R) was registered by visual check of thoracic movements. In case of bradycardia (less than 100 beats per minute) and/or hypotension (mean arterial pressure < 60 mmHg) and/or hypoventilation (<10 breath acts per minute), atipamezole IM (Revertor; Cp Pharma Mbh, Burgdorf, Germany) was administered at 2.5x previous dexmedetomidine dose and flumazenil (Anexate; Help a Pharm GmbH, Hamelin, Germany) 0.02 mg kg^−1^ to reverse dexmedetomidine and midazolam side effects, respectively. In case of apnea, trachea was intubated with PVC endotracheal tube (Rusch; The Sheridan, Morrisville, NC, USA) after 2% of lidocaine splash in the arytenoids (0.2 mL cat). The endotracheal tube was connected to a pediatric T piece and manually ventilated (6 acts per minute aiming to ensure an end tidal of CO_2_ between 28 and 35 mmHg) until spontaneous breathing was returned. Rectal temperature was measured every 10 min using digital thermometer (MT4233; Sejoy Electronics & Instruments Co., Hangzhou, China).

### 2.3. Statistical Analyses

Sample size calculation was performed using an ANOVA Two-Way model for repeated test with power of 0.8, *α* of 0.05, and considering mean and SD of the decrease in tear flow in a previous study [23] (G*Power version 3.1.9.6; University of Düsseldorf). A minimum simple of 9 cats per group was calculated. A total number of 20 cats were recruited and divided into two groups.

The data referring to the tear secretion were checked for normality with the D’Agostino and Pearson test and were then processed with an ANOVA Two-Way model for repeated measures (https://www.statskingdom.com/two-way-anova-calculator.html, URL accessed on 15 June 2022). The Tukey–Kramer test was used post hoc at different times in each group. The data referring to the STT-I reaction were processed with Chi-Square test. Through the manuscript, if not otherwise indicated, the data are presented as mean ± standard error. The differences are considered significant for *p* < 0.05.

## 3. Results

A total of 20 European shorthair cats were included and evenly allocated into two groups. The demographic data and ASA status classification do not present statistical differences (Table 2).

Tear production was taken into account and analyzed for the eyes that were assessed first, which were the right eye of each cat. Tear secretion in the IR group, analyzed for repeated measures at each time point, was not significatively different (*p* = 0.1477); when we analyzed the data referring to the right eye with a model that takes into account, using as a variability source the time (T0-T6), the method of administration (IM vs. IR) and the interaction between time and method of administration, the result was significantly different, with a marked tear reduction in the IM group compared with the IR group (*p* < 0.001) (see Figure 1).

In terms of reaction to STT-I at T0, this does not present significant differences in each group between right and left eyes. Significant differences were found at T2 at scores 0 and 1 (*p* = 0.005 and 0.0007, respectively) and at T3 at score 0 (*p* = 0.02) in the right eye.

In the left eyes between groups, statistically significant differences were found at time points T4 and T6. In particular, T4 was significative for score 0 (*p* = 0.02), and T6 was significative for scores 1 and 2 (*p* = 0.003 for both scores). For each time point, results are shown in Table 3.

No complications such as bradycardia, hypotension and/or hypoventilation were detected during the study in both groups. No cats received atipamezole and/or flumazenil.

## 4. Discussion

Based on the findings of this study, the results suggested a decrease in tear production during sedation with a mixture of dexmedetomidine, ketamine and midazolam in both groups of cats. The tear production decrease is statistically significant only in the IM group.

Similar results were obtained in previous studies regarding the use of sedative agents in healthy cats [23,25]. To our knowledge, these results are not comparable within the literature, as there are currently no other studies that have evaluated the clinical impact of IR administration of sedative agents on tear production.

Physiological tear production in cats is extremely variable. In healthy cats, Rajaei and colleagues [27] showed a mean tear production of 14.9 ± 4.8 mm/min. Similar values were described by Cullen and colleagues [28], with 17.4 ± 4.6 mm/min of mean and standard deviation. A decreased tear production can lead to various ophthalmic pathologies, such as keratoconjunctivitis sicca [29]. Uhl et al. [30] have reported values < 9 mm/min in cases of conjunctivitis, corneal ulceration, non-ulcerative keratitis, symblepharon and eosinophilic keratitis. In this regard, the cats included in the study were examined by a veterinary ophthalmologist to ensure that one or more of these diseases were not present and changed the results. Also, demographic data such as sex, age, weight, neutering and spaying have been studied as factors that can affect tear production [27,31,32]. In our study, demographic data are not considered influencing factors, as they are not statistically significant between the groups. Other causes of decreased tear production may be related to changes in certain ocular variables during anesthesia, such as the central position of the eye, the lack of the blink reflex, the reduction in corneal sensitivity or the impairment of the tear reflex [33]. In addition, the alteration of the cellular response to stimuli by tear cells [34,35], the central effect of sedative agents on the nervous system (CNS), vasoconstriction, antinociceptive effects and an altered metabolism of the gland’s cells may be other factors influencing tear production [24,25,26].

### 4.1. The Role of Anesthetic Agents on Tear Flow

The use of several sedative drugs has been observed to reduce tear secretion, more markedly even in a short-term period [23,25]. Usually, this reduction in production is transient, as was also demonstrated by Peche and colleagues [36], where tear hypoproduction in cats was observed from 6 to 18 h after the end of general anesthesia. How anesthetics interact with tear production has also not been clear in dogs and cats. In the feline species, the effect of acepromazine and xylazine as depressing agents on tear production was demonstrated by Ghaffari et al. [23] for the IM route. In a more recent study, Di Pietro et al. [25] showed that cats undergoing castration or ovariectomy who were premedicated via the IM route with medetomidine and ketamine had a considerable decrease in their tear film production. These studies use a drug protocol that involve alpha-two agonists (xylazine and medetomidine, respectively). In our study, another alpha-two agonist, such as dexmedetomidine, is proposed. Ghaffari et al. [23] used xylazine 2 mg kg^−1^ and 0.2 mg kg^−1^ acepromazine via the IM route as the only drug administered to each group. The pre-administration values (T0) obtained in the xylazine-treated group differed from our results. The values presented by Ghaffari et al. [23] are 13.93 ± 1.18 mm/min.; these are higher than our values of 9.9 ± 4.6 mm/min. Post-administration values are more complex to interpret, as the timing of data collection is different. However, in the study by Ghaffari et al. [23], a lower value of tear production of 2.18 ± 0.97 mm/min was reached at 15th minute post-administration; in our study, the lowest absolute value was 4.5 ± 2.01 at the 20th minute. In contrast, Di Pietro et al. [25] proposed a mixture of medetomidine of 0.08 mg kg^−1^ and ketamine of 5 mg kg via IM. In our study, dexmedetomidine is used at 0.003 mg kg^−1^, a significantly lower dosage compared with the aforementioned studies. The different use of the alpha-two appears to elicit a more pronounced clinical response in terms of tear production and duration of effect [37]. The alpha-2 agonists show a dose-dependent clinical effect, particularly with regard to cardiovascular, respiratory and metabolic effects, where a more pronounced effect is manifested as the dose increases until a ceiling effect is reached [37,38,39]. Comparing studies from a clinical response point of view is complicated. In any case, the studies agree on the results (decrease in tear production), although the variability between xylazine, medetomidine and dexmedetomidine is marked. For this reason, it is conceivable that alpha-two agonists play a significant role in tear hypoproduction. The hypothesis for the reduced tear production values can be traced back to the effect of alpha-two agonists on the cardiovascular system. Alpha-2 provokes dose-dependent cardiovascular effects, which result from the activation of the baroreceptors, followed by arterial hypertension due to peripheral vasoconstriction [37,38]. The centralization of the blood flow causes a reduction in the oxygenation of peripheral organs such as the lacrimal glands, resulting in reduced physiological activity. A further cause, reported by Leonardi et al. [26] in dogs, could be associated with the postsynaptic activation of alpha-adrenergic receptors at the level of the central nervous system.

The role of ketamine as a *solo* anesthetic agent on tear production was not investigated. Arnett et al. [40] evaluated the decreased tear production caused by a ketamine, acepromazine and atropine (an anticholinergic agent that provokes hypolacrimia) mixture in cats, and it is not clear if ketamine affects tear production directly. The values reported in Di Pietro et al. [25] deviate severely from our results, probably due to the substantial difference in proposed drug dosage. The ketamine dosage proposed in our study is less than half for the IM route of administration (2 mg kg^−1^ of ketamine). However, in the authors’ opinion, according to Clanachan and colleagues [41], a possible cause of decreased tear production could be attributed to the central positioning of the eyeball and the increased sympathetic tone that ketamine induced.

No studies are available for the effect of benzodiazepines on tear production in cats. Using the rabbit as an animal study model, Ghaffari et al. [42] evaluated the impact of a single administration of a phenothiazine (acepromazine) and a benzodiazepine (diazepam) on tear production. The acepromazine and diazepam were administered at the same dosage of 1 mg kg^−1^ via the IM route. The study shows that significantly lower STT-I values at the same time points, by more than 50 per cent, were found in rabbits who received acepromazine. The same study also shows that the STT-I values of rabbits that received diazepam at T0 are overlapping at T15 and T25 post-administration. Based on this study, it is reasonable to assume that benzodiazepines do not significantly affect tear production. Given the slight cardiovascular impact of midazolam on the cats [43], it is reasonable that benzodiazepines have a mild effect on tear film production in cats as well as rabbits. However, this hypothesis is speculative. In addition, the combination of drugs with a synergistic effect [44] could also have this effect on decreased tear production. The pharmacokinetic reason for the lower impact of the IR route of administration compared to the IM route is currently unclear. To the author’s knowledge, there is only one clinical study in which an alpha-2 agonist is used for the IR route combined with midazolam and ketamine [17]. The study demonstrates how there is clinical efficacy via the IR route with a lower cardiorespiratory impact compared to the IM route. The pharmacokinetic reason is not entirely clear but this, from the results obtained in this study, would also seem to influence tear film production less.

### 4.2. Tear Film Production

Both groups demonstrated a reduction in tear film production. Regarding the variation over time in tear flow, the right eye results were statistically significant.

The data collected on the left eye were not taken into consideration at the end of the statistical analysis. This could have resulted in increasingly higher “false values” in the left eye than in the right one. There may be different mechanisms: (1) the positioning of the paper strip causes an increase in the reflex secretion; (2) a mild activation of a vagal reflex by compression on the optic nerve with parasympathetic activation of the autonomic system causing an increase in tear secretion [22,45].

As Figure 1 shows, the reduced tear production is more pronounced in the IM group than in the IR group. The IM group exhibits a sharp drop already at T2 and a minimum peak at T4, with values of 3.5 ± 2.0 mm/min vs. 9.1 ± 6.9 mm/min in the IR group. The IM tear values between our study and Ghaffari et al. 2010 [23] and Di Pietro et al. [25] are in agreement. The designs of the studies are different in terms of the animals involved, dosages, timing of STT-I detection and proposed drug mixture. The difference between the IM and IR group values can be attributed to the reduced clinical impact that the IR route has compared to others [46]. In fact, the lowest value recorded in the IR group is at T5 (8.6 ± 4.8 mm/min) and remains significantly higher than the lowest peak observed in the IM group (3.5 ± 1.6 mm/min). This observation is also highlighted in Figure 1, where the trend of IR tearing is more stable, unlike the “up and down” trend of the IM group.

Although not statistically significant, the values at T6 between the two groups are relevant. In fact, IM values are lower than IR, as well as being lower than the baseline (T0) of the IM group itself. This leads us to deduce that the clinical effect on tear production of the IM group does not end at the end of the sedation but that a tail of the effect lasts, as reported by Peche et al. [36], in which tear values seem to return to baseline levels only after 6–18 h post drug administration. In contrast, the T0 and T6 values of the IR group are overlapping, which suggests that the recovery of the physiological tear production in the IR group is faster.

### 4.3. STT-I Reaction

The trends in the data obtained differ from the initial hypothesis. Significant results were found relating to the STT-I reaction at certain time points. The IM group presented a smaller STT-I reaction than IR group in both eyes. At T0, the reaction to STT-I showed no significant differences between the two groups. The STT-I reaction is significative in the right eye (the first to be assessed in all cats for all measurements) at T2 and T3; otherwise, it is significative in the left eye at T4 and T6. The IR route is characterized by high bioavailability and a rapid onset [46]. Therefore, the IR group could have had a reduced response to STT-I at earlier time points than the IM group, but this trend is not supported by our results. In this case, the data are not clearly understood, and the hypotheses underlying this pharmacokinetic behavior may be different. For the IR route, the pharmacokinetics of dissociative agents and benzodiazepines have been investigated in veterinary medicine [47,48], which is not so for alpha-2 agonists. In contrast, the pharmacokinetics of alpha-2 agonists have been extensively studied for the IM route [37]. Alpha-2 agonists possess the characteristic of having a dose-dependent onset [37], and this could also be a reasonable behavior for the IR route, but this is not supported by the literature. With the same dose of dexmedetomidine administered, the IM group has a smaller reaction to STT-I from the first time points. The dosage of dexmedetomidine proposed for the IR route is a low dose, taken into account in order to have a mild/moderate sedation to allow small clinical/diagnostic procedures. Indeed, all cats in the study completed their procedures without requiring extra sedation or deepening of the anesthesia plan. The proposal of a higher dosage for IR administration could have sped up the onset. Given the fact that even at the later time points, the IM group reacted less, it is possible to assume a longer tail of drug action for this route.

This study has some limitations. Based on a lack of research on IR drug administration, as regards dexmedetomidine, choosing an optimal dose for the study design was a challenge. If drugs were used at higher doses, the clinical effects on cats would have been more marked; at the same time, we could also have side effects that had to be avoided. A pharmacokinetic study on the use of dexmedetomidine by the IR route is desirable. In any case, the proposed protocol is used to ensure short-term sedation for rapid procedures that require chemical immobilization with mild to moderate analgesic coverage. The number of animals involved in this study is limited, although this resulted from a power calculation.

Another limitation is the state of sedation between the two groups. Indeed, a greater tear film decrease in the IM group could be attributable to a higher state of sedation. The sedation status in this study was not assessed, so this cannot be determined. However, the sedation generated by the drugs was enough to complete the procedures for which chemical immobilization had been requested. Another main bias of the study is that repeated measurements between the right and left eyes for each measurement may have influenced the second measurement. In any case, the interest of the authors is not in the absolute value but the trend over time of the IR route, which is not influenced by this type of measurement.

## 5. Conclusions

The use of dexmedetomidine, ketamine and midazolam via the IR route appears to have less impact on tear production than in the IM group. At the proposed dosages, the IR route appears to have a negligible impact on tear production. Pharmacokinetic studies for alpha-2 agonists, dissociative agents and benzodiazepines for the IR route are recommended to confirm these findings. The data available also recommended the use of eye gel in cats receiving sedation via the IM or IR route. Further investigations are needed.

## Figures and Tables

**Figure 1 animals-14-00145-f001:**
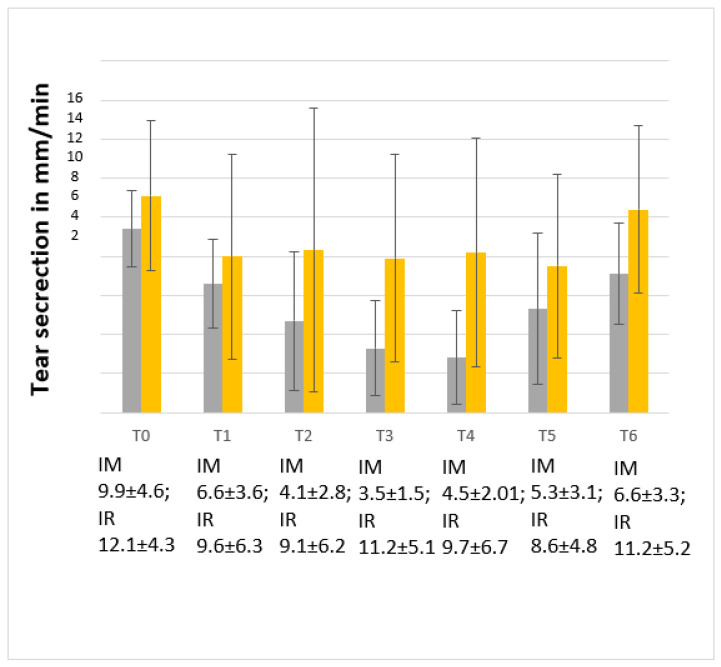
Tear flow in the right eye over time during sedation in the two groups. The first histogram refers to the IM group, the second to the IR group. On the y-axis line, tear secretion is expressed as mm/min. On the x-axis line, time is expressed as T0 (1 h before drug administration), T1 (2′), T2 (10′), T3 (20′), T4 (30′), T5 (40′) and T6 (80′). Tear production is reported as media and standard deviation. In the IR group, tear film production is higher than in the IM group.

**Table 1 animals-14-00145-t001:** Assessment score of STT-I strip placement. The score varies from a value of 0 to 3 depending on the cat’s reaction. The response to strip placement was performed by the same operator for all cats involved in the study.

Assessment Score and Definition
0: Test execution without restraint of the cat. No stiffness, vocalizations or shirk from strip positioning
1: Test execution with mild restraint of the cat. No stiffness, vocalizations or shirk from strip positioning
2: Test execution with restraint of the cat. Presence of stiffness and vocalizations but no shirk from strip positioning
3: Test execution with restraint of the cat. Persisting presence of stiffness, vocalizations and shirk from strip positioning

**Table 2 animals-14-00145-t002:** Demographic data and ASA status classification. Age and weight values are expressed as mean and standard deviation. Simple sex, population split by entire male (EM); neutered male (NM); spayed female (SF); entire female (EF). ASA status population split by ASA I or II.

	IM Group				IR Group				*p* Value
Age (years)	5.5 ± 2.50				5 ± 3.50				0.09
Weight (kg)	4.78 ± 1.89				5.5 ± 2.02				0.38
Sex (neutered or spayed)	EM	NM	SF	EF	EM	NM	FS	EF	0.66
	4	4	2	0	2	4	4	0	
ASA status (I or II)	I		II		I		II		0.89
	8		2		6		4		

**Table 3 animals-14-00145-t003:** Evaluation of the strip reaction for STT-I at different time points, analyzed by Chi-Square test. The table shows the right (R) and left (L) eye values among IM and IR groups. The degree of reaction is expressed as a number from 0 to 3. Time points were 1 h before drug administration, (T0) 2′ (T1), 10′ (T2), 20′ (T3), 30′ (T4), 40′ (T5) and 80′ minute post administration (T6). For each right and left eye score, there are *p* values below.

	Score 0		Score 1		Score 2		Score 3	
	R	L	R	L	R	L	R	L
I M-T0	0	0	4	0	4	5	2	5
IR-T0	0	0	2	0	7	2	1	2
*p* value	1	1	0.62	1	0.37	0.35	1	0.35
IM-T1	0	0	5	2	3	3	2	5
IR-T1	0	0	8	2	1	5	1	3
*p* value	1	1	0.35	1	0.53	0.65	1	0.65
IM-T2	8 5		0	4	2	1	0	0
IR-T2	1 1		8	4	1	5	0	0
*p* value	0.0055	0.14	0.0007	1	1	0.14	1	1
IM-T3	9	7	1	3	0	0	0	0
IR-T3	3	3	5	4	2	3	0	0
*p* value	0.02	0.18	0.15	1	0.47	0.21	1	1
IM-T4	8	7	2	3	0	0	0	0
IR-T4	3	1	4	7	3	2	0	0
*p* value	0.07	0.02	0.062	0.18	0.21	0.47	1	1
IM-T5	2	3	8	7	0	0	0	0
IR-T5	2	0	7	6	1	4	0	0
*p* value	1	0.21	1	1	1	0.08	1	1
IM-T6	1	0	8	10	1	0	0	0
IR-T6	0	0	9	3	1	7	0	0
*p* value	1	1	1	0.003	1	0.003	1	1

## Data Availability

The data generated in this study are presented in the tables of this article. For any further information, the reader can contact the authors.

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
