# Peer review of "A Comparison of the Intrarectal and Intramuscular Effects of a Dexmedetomidine, Ketamine and Midazolam Mixture on Tear Production in Cats: A Randomized Controlled Trial"

_animals, 2023, doi:10.3390/ani14010145_

Round 1

Reviewer 1 Report

Comments and Suggestions for Authors

The study results are sound and consistent.

As authors stated one difference in regard of tear flow may be different levels

of sedation. Are the authors able to present any seadtion scores that may (semi)quantify this effect? Otherwise they shall not only dicuss this pehnomenon (eg l 300ff) but may present it as a clear limitation.

Further were there any differences in regard of treatment? E.g. more painful treatment may explain less sedation and more tear flow?

On which base did the authors decided to include only ten animals per group? This small number may be added as a limitation too.

If IR sedation may have comparable effects on allowing painful therapies, the advantages in regard of side effects such as avoiding the risk of muscle tumors at the injection sides this should be discussed more clearly.

Line 103: There is an unnessecary fullstop in the middle of the sentence.

Author Response

Dear reviewer,

Thank you very much for your comments.

Attached is the word file with our corrections

Reviewer 2 Report

Comments and Suggestions for Authors

Dear author,

Congratulations for the study. The objective of this study is interesting but I think that it needs important changes before publication. Also the English language need to be improved.

I have included some corrections that I hope can help you to improve the quality of the manuscript.

The objective of this paper is to assess the impact on tear film production of the intrarectal route administration of a mixture of dexmedetomidine, ketamine and midazolam in comparison with the intramuscular route administration in cats.

The reaction to the Shirmer tear test I (STT-I) placement was evaluated but it is not included as an objective of the study.

Hypothesis:

-IR administration has a reduced impact on tear production compared to the IM route.

-IR route allows a faster recovery of tear basal values than the IM route.

INTRODUCTION:

-Line 74. “The aim of the study is to compare tear production after IR and IM sedation in cats”. Please include “…sedation with a mixture of dexmedetomidine, ketamine and midazolam in cats”

MATERIALS AND METHODS

-Line 82. I suggest to include the clinical procedures that were performed, since it can help to the reader to understand “the average level of sedation”

-Line 102. Please change anesthetics for drugs

-Line107. When did you perform the ophthalmological examination? Please include it.

How do you think that eyes manipulation as evaluation of the intraocular pressure or eye staining with fluorescein could influence the tear film production? Baseline values? Please discuss it

-Line 115. What is the objective of evaluate the reaction to STTI placement?  How does it contribute to your study “comparison of intrarectal and intramuscular effects of dexmedetomidine, ketamine and midazolam mixture on tear production in cats: A randomized controlled trial??

Do you want to evaluate sedation quality? Please, clarify it.

Line 117. The score varies from 0 to 3. Please correct it.

-Line 124. Please clarify the cardiovacular parameters recorded

-Statistical analyses: Why did you included 20 cats? You must explain how did you calculate the sample size. It is important to validate the results.

RESULTS

-Lines 153-157. The paragraph is not clear. “No significant differences were found at T0 to right and left eyes in each group”. .. respect to what?? Tear production, STT-I reaction?? (since you show values of score, I understand that it is STT-I reaction, but it is nos explained) Also it should be said …at T0 between right and left eyes…

The second sentence I guess you mean “ in term of STT-I reaction in the right and left eyes between IM and IR groups..”

Please clarify the entire paragraph.

-Line 158. Remove RIGHT

-Table 3. I would recommend to eliminate “other score” since it doesn´t add information and make more difficult to understand the results. Include just the number of animals out of 10 included in score 0, score 1, score 2 and score 3.

Once you remove “other scores” you can unify table 3 and table 4. I think 2 tables (one per eye) are not necessary to show the result of a chi-square test of a parameter that is not the objective of the study

-Line 162 and line 172. The score is 0-3 and not 0-4

-Lines 165-167. You have to improve the english language to make it understandable

-Line 168. Please remove left

-Lines 174-179.  The objective of the study is to evaluate the tear film production, so it is the most relevant result and it should be the first result. Please include it before the STT-I production result.

The reaction to the Shirmer tear test I (STT-I) placement was evaluated but it is not included as an objective of the study.

            -Line 174-175. Please clarify if in “tear secretion in IR group” you include right and left eyes.

            -Please consider to include another graph to show the tear production in the left eye.

-Graph 1.. Values of tear film production in IR group are no included neither in the graph or in the results section. This is a big mistake, since it is the objective of the study.

-Line 185.Results of the IR group have to be showed with numerical data and including the p value in the graph. Remove the sentence, since you are not given statistical results.

-Please include that none cat received atipamezole.

DISCUSSION

-Line 187-189. Please make clear that the tear production decreased in both groups respect to baseline values, but it decreased significantly only in the IM group.

-Line 190. “The results of our study also confirm that…” It is not an objective of your study, and do not included it as an objective and don’t include a hypothesis on STT-I placement reaction.

-Please explain how sex is a negative factor on tear production.

-Line 217. Paper 36 showed that tear hypoproduction in cats was observed from 6 to 18h after the end of anesthesia. Why did you evaluate only during 80 minutes after drug administration?

-Line 238-245. Please eliminate it since it doesn´t contribute to the discussion. You can use the Di Pietro study to discuss related the lowest tear production since they used medetomidine and ketamine (similar to your drugs mixture), but the rest of the information is not necessary.

-Line 255. Please correct the hypothesis for the reduced  STT-I values to reduced tear film production

-Line 292-293. Not understandable

-Line 295. What do you mean with “less pronounced clinical effect”?? You do not include in material and methods section other evaluation than tear production and STT-I reaction. Please clarify it.

-Line 306. You do not evaluate clinical sedation level. Please remove it.

-Lines 295-307. Please re-write the paragraph, since it is difficult to understand. The english language has to be improved. You are mixing STT-I reaction with sedation level (and it is not the same) eg. The sentence “ the stimulus of the STT-I was significant respect to the sedation status in IR group”. You did not evaluate sedation!

Please make clear the literature relating IR route (since you say that it is reduced only for pharmacokinetic studies of alpha-2 agonist).

-Line 311. Your second hypothesis is that a higher dose of dexmedetomidine in the IR group probably could have had marked clinical effects and more lasting. It is not a plausible option to explain that the cats in the IM group had a lower score for STT-I reaction. Please clarify it.

-Line 318. Please clarify “statistically significant”. In the results section you said that tear secretion in IR group was not significantly different (line 174)

-Paragraph line316-343. In the same way than the results section, this is the objective of the study and it should be discussed at the beginning of the discussion. Actually information related to tear production studies is repeated.

-Line 331. Please discuss this sentence deeper. What does clinical impact means?

-Line 337-343. Your second hypothesis is “IR route allows a faster recovery of tear basal values than the IM route”. Since it is part of your objective, I would suggest to go deeper on it.

-Line 344. Why did you increase the dose of ketamine and midazolam in the IR group? I haven´t found an explanation along the paper. The explanation on limitations of the study is not enough. You include a big section “Role of anesthetic agents on tear flow” discussing on drugs effect. I think you have to justify in this section the different doses used in both groups.

-Line 350. Please correct it since you did not evaluate level of sedation. It have to be modify from material and methods, since you said STT-I reaction.

CONCLUSIONS

-Line 361.Please change anesthetics for dexmedetomidine, ketamine, midazolam mixture at the studied doses.

Also change the order: IR administration has lower effect on tear production, since the objective is evaluate the IR route.

-Line 362. You have not evaluated if the IR route (dexmedetomidine, ketamine midazolam; not in general) can be used for the execution of short clinical procedures, so it can not be part of the conclusion.

-Line 364. Please clarify what “cats receiving infrared sedation” means

Comments on the Quality of English Language

Quality of english need to be improved

Author Response

Dear reviewer,

The authors are grateful for corrections and suggestions. Below is the file with the corrections.

The authors remain available for any clarification

Reviewer 3 Report

Comments and Suggestions for Authors

Tear secretion is a highly debated topic in the field of ophthalmic anesthesia. The study was conducted very well, congratulations to the authors. Alpha2 agonists reduce tear secretion, but are very suitable for eye surgery. We therefore suggest that the authors add and comment on the following bibliographical note:

Giovanna L. Costa, Fabio Leonardi, Claudia Interlandi, Filippo Spadola, Fisichella Sheila Francesco Macrì, Bernadette Nastasi, Daniele Macrì, Vincenzo Ferrantelli, Simona Di Pietro (2023). Levobupivacaine combined with cisatracurium in peribulbar anaesthesia in cats undergoing corneal and lens surgery. ANIMALS, p. 00, ISSN: 2076-2615, doi: 10.3390/ani13010170

Author Response

Dear reviewer,

The authors are grateful for the corrections made.

Round 2

Reviewer 2 Report

Comments and Suggestions for Authors

Dear authors. Thank you for the changes that you have made on this paper. It improved much but I think the paper still needs some changes before being published. You have re-written the objectives and eliminated all the information related to sedation (since you do not include evaluation of sedation in your methodology) and moved it to reaction to the STT-I . I appreciate the effort to change it but I don´t think that the reaction to the STT-I  can add any valuable information to the reader, since you have not stablished a relation between tear production and reaction to STT-I in your methodology. For this reason, I would recommend this work to be published as a short communication with the data of the tear production only.

Comments on the Quality of English Language

English language needs to be improved. I don´t think the quality of the language is good enough for publication.

Author Response

Dear reviewer,

Authors thank you for the review process you are carrying out. All your comments and corrections are making significant improvements to the quality of the paper.

For the table correction and the other comments, we have tried to make the changes you requested.

Concerning the evaluation of the Shirmer T test the authors feel that we have not sufficiently explained our point of view and they will try to be clearer below.

The point you make about the correlation between sedation and tear production is a fair and interesting observation, but it is not the focus of the paper. Data collected and processed in the previous paper (https://doi.org/10.3390/vetsci9100520) evaluate the sedation status of the two alternative routes of administration and comparing them (IM and IR). The focus of that paper, however, is not solely on the assessment of sedation status. That paper is a clinical trial that always predicts sufficient sedation efficacy to perform the required procedures. The drugs and dosages were chosen according to the following criteria: “the lowest that would give a sufficient clinical effect to guarantee the procedure'”. As reported in the paper, this included cats of a quiet disposition presenting to the teaching hospital to perform short procedures and of low algic stimuli. This is why, together with an unfamiliar route of administration such as IR, we decided on dosages that tended to be low. In fact, we also evaluated the cardiorespiratory impact that the two routes of administration at the proposed dosage gave. Our hypothesis was to provide sufficient sedation for the procedure with less cardiorespiratory impact. In our published results, this occurs with a lower state of sedation in the IR. As specified, all procedures were completed without difficulty, so the fact that IR sedation at the proposed dosages gave less sedation is an important finding that indicates to us that certain procedures can also be performed administering low dosage dexmedetomidine, ketamine and midazolam via IR.

The idea is also applicable to the evaluation of tear production and the evaluation of the Shirmer T test. In fact, our hypothesis was to have tear production for the IR route preserved compared to the IM route. The evaluation of the Shirmer T test was introduced to evaluate whether and how much the stimulus of the positioning of the strip could influence tear production. From the results presented, both IR and IM appear to interact in a limited manner, regardless of the state of sedation of the cats examined.

We can talk about it in discussions by mentioning the article, but we cannot include the results of another article and compare them.

In any case, the possibility of not presenting this data because it is considered non-influential and irrelevant puts us in difficulty as this idea contrasts with that of the other 2 reviewers. Even if we did not present this data with the green light of the other two reviewers, we do not see why the paper should be presented as a short communication. In fact, the papers cited on the evaluation of tear production by Ghaffari and Di Pietro were published as articles randomized controlled trial.

Line 129: “arterial” was added;

Table 3: The left eye score 3 values are present in the table, probably the layout of the table has moved when loading the word file for this reason it seems to be missing or is confusing. The authors reloaded the table differently to try not to have layout problems again. As reported in the statistical analysis where not reported, significance is set at p<0.05. A sentence has been added to specify the presence of the p value for each right and left eye score.

Graph 1 the table has been checked and modified; the variation of the curve over time is significant and not the single value at each time point as described in the results;

Line 208: “in cats” added;

In the first round of revisions, whether or not gender was a factor in tear production was corrected. It actually clarifies the fact that it has been studied and the fact that it does not appear to be involved as a factor. The sentence on line 223-224 says: “Also demographic data such as sex, age, weight, neutering and spaying have been studied as factors that can affect tear production”. From this sentence it is clear that these factors have been studied as potential factors that can influence the tear film and not that they necessarily influence it or not. From our results, we did not find any significant differences therefore the data are shared specifying that the authors did not highlight any significant differences. In line 245 There was a typo left over from the first round of corrections, which is why we proceeded to remove it. Therefore, step by step we inserted all the demographic and non-demographic factors that were studied to see whether they influenced tear production or not. Those mentioned except sex seem to influence, at least in part and also with a mechanism that is not entirely clear but our results are not significant in this sense.

Line 221-227 and 243-245 As suggested, the concept is repetitive, which is why it has been eliminated.

Line 228: discussing why IR sedation should have less impact on tear film is not clear. The first reason is linked with a few studies on pharmacokinetics and dynamics in cats for this route and it is complex to hypothesize just from the results of this study. The pharmacokinetic effect probably needs to be interpreted on the behavior of alpha-2 agonists, for which there is currently no pharmacokinetics of the IR route. In any case, given the suggestion provided to us, the authors have provided food for thought in this sense at the end of the paragraph;

Line 234-236 added;

Line 294-301: A reported in the results: ”Tear production was taken into account and analyzed for the eye assessed first, then all the right eyes of each cat.” only the first eye examined for the evaluation of the variation over time was analyzed. The discussions seem to deviate from what the authors' concept is, for this reason we have made some changes in order to be clearer on the data analyzed and presented;

The two paragraphs below have been modified according to the authors' idea to go against the reviewer's suggestion. As regards the interest of alternative administration routes, the authors believe they have briefly but comprehensively explained why an alternative route such as IR compared to the two conventional routes such as IV and IM can be useful. The growing interest is demonstrated by several recent publications in these routes such as IR, IN and OTM in dogs and cats. The focus of the paper, however, is not to demonstrate or invite the reader to use an alternative route of administration to the detriment of the canonical ones. In the opinion of the authors, the reader does not need to have explanations regarding the interest of this route but must become aware that this alternative administration, if chosen, could have a reduced impact on the tear film.

Hoping everything is clear, the authors remain at your disposal. We look forward to hearing from you soon, we wish you good work.

Best regards